# Important Cells and Factors from Tumor Microenvironment Participated in Perineural Invasion

**DOI:** 10.3390/cancers15051360

**Published:** 2023-02-21

**Authors:** Zirong Chen, Yan Fang, Weihong Jiang

**Affiliations:** 1Department of Otolaryngology-Head and Neck Surgery, Xiangya Hospital, Central South University, Changsha 410008, China; 2Otolaryngology Major Disease Research Key Laboratory of Hunan Province, Changsha 410008, China; 3National Clinical Research Center for Geriatric Disorders, Xiangya Hospital, Central South University, Changsha 410008, China; 4Anatomy Laboratory of Division of Nose and Cranial Base, Clinical Anatomy Center of Xiangya Hospital, Central South University, Changsha 410008, China

**Keywords:** perineural invasion (PNI), tumor microenvironment (TME), single-cell spatial transcriptomics (sc-ST)

## Abstract

**Simple Summary:**

Perineural invasion (PNI) as the fourth way for solid tumor metastasis and invasion has attracted a lot of attention, and axon growth and possible nerve “invasion” to tumors has also become an important component of PNI. We aim to summarize the current theories on the molecular mediators and pathogenesis of PNI, add the latest scientific research progress, and especially explore the use of single-cell spatial transcriptomics in this invasion way, to obtain a better understanding of PNI.

**Abstract:**

Perineural invasion (PNI) as the fourth way for solid tumors metastasis and invasion has attracted a lot of attention, recent research reported a new point that PNI starts to include axon growth and possible nerve “invasion” to tumors as the component. More and more tumor–nerve crosstalk has been explored to explain the internal mechanism for tumor microenvironment (TME) of some types of tumors tends to observe nerve infiltration. As is well known, the interaction of tumor cells, peripheral blood vessels, extracellular matrix, other non-malignant cells, and signal molecules in TME plays a key role in the occurrence, development, and metastasis of cancer, as to the occurrence and development of PNI. We aim to summarize the current theories on the molecular mediators and pathogenesis of PNI, add the latest scientific research progress, and explore the use of single-cell spatial transcriptomics in this invasion way. A better understanding of PNI may help to understand tumor metastasis and recurrence and will be beneficial for improving staging strategies, new treatment methods, and even paradigm shifts in our treatment of patients.

## 1. Background

There are three well-known traditional ways for solid tumor metastasis and invasion: direct invasion of surrounding tissue, lymphatic spread, and haematogenic spread [1]. With the deepening of tumor research, the fourth invasion and metastasis mode has gradually attracted the attention of researchers: perineural invasion (PNI). In the process of tumorigenesis, nerves will be induced to sprout into the tumor during tumorigenesis, which in turn leads to cancer–nerve crosstalk, and assists tumor dissemination [1,2,3,4,5]. PNI was first reported by European scientists who described head and neck cancers that show a tendency to invade along nerves when moving to the intracranial fossa [6,7], which is usually associated with poor prognosis and may lead to severe pain, typically pain such as pancreatic adenocarcinoma (PAAD) [1,4,5,8]. Since then, PNI has become the key pathological feature of many malignant tumors, such as PAAD [8], prostate cancer (PRAD) [9,10], colorectal cancer [11], squamous cell carcinoma (SCC) of head and neck mucosa [4,5,12,13,14,15], adenoid cystic carcinoma (ACC) [2,16], etc.

As a unique pathological entity, PNI can be observed without lymphatic or vascular invasion. Nerve sheath was composed of three connective tissue layers, which are the epineurium, the perineurium, and the endoneurium [1]. The endoneurium is the innermost layer formed mainly by axons and Schwann cells, and also contains mast cells, resident macrophages, fibroblasts, and blood vessels [17]. A laminated cylindrical layer derived from fibroblasts formed the perineurium to surround the endocardium, and they connected with each other through tight connections and gap connections to form a layer separated by collagen fibers (type IV) [18]. The epineurium is the outermost layer and includes a collagen tissue sheath, a plexus of blood vessels, lymphatic vessels, resident macrophages, fibroblasts, mast cells, and sometimes adipose tissue [19,20]. The definition of PNI was based on the nerve sheath: the tumor is close to the nerve and involves at least 33% of the circumference of adjacent nerve sheaths, or cancer cells exist in any of the three layers of adjacent nerve sheaths [1]. The incidence of PNI is different among different cancers, such as 70–100% in PAAD [8], 85–100% in PRAD [21], 20–57% in colorectal cancer [11], 25% to 80% in SCC of head and neck mucosa [4], and up to 80% in salivary adenoid cystic carcinoma (SACC) [22]. Cases reported of PNI were commonly associated with poor prognosis and decreased survival [1,11,23,24]. It has been reported that when PNI occurs in nerves with a diameter of >1 mm, there is an independent correlation between a higher local recurrence rate and PNI [25]. It has also been proved that the difference in the position of the PNI site relative to the tumor will also lead to the difference in patient prognosis [26]. Therefore, some researchers use the term “nerve invasion (NI)” instead of “perineural invasion (PNI)” to describe this tumor behavior. Given the importance of the invasion mode and the extremely serious clinical consequences, researches on this point are essential to control tumor dissemination and monitor disease progression.

Although most research focused on the invasion of tumor cells to nerves, recent research reported that PNI starts to include axon growth and possible nerve “invasion” to tumors [27]. More studies have recognized PNI as the result of the interaction of tumor cells, nerve cells, and nerve microenvironment, and all parties work together to promote the invasion of tumors into the space around nerves [16]. The tumor microenvironment (TME) plays an important role in PNI. The interaction of tumor cells, peripheral blood vessels, extracellular matrix (ECM), other non-malignant cells [28,29], and signal molecules in TME play key roles in the occurrence, development, and metastasis of cancer [2,22,30,31,32,33]. The nerve barrier constitutes a defense line against tumor invasion. When tumor cells invade, peripheral nerve cells release neurotrophic factors, growth factors, chemokines, and cell-adhesion molecules to promote nerve repair and regeneration and maintain the stability of barrier function. At the same time, nerve cells such as Schwann cells and tumor cells interact with each other through autocrine and paracrine signals, enhancing the adhesion between cancer cells and nerves, and promoting tumor diffusion along nerves [34]. All kinds of cells in the microenvironment, including tumor-associated macrophages and fibroblasts, are activated by chemokines to produce peripheral inflammation, release neurotrophic factors, growth factors, and MMPs, and promote the growth, proliferation, and invasion of tumors, resulting in PNI [34,35,36]. As our understanding of the pathogenesis of PNI continues to progress, so will the definition of PNI [37,38]. The study of TME promotes the understanding of the existence and function of nerves in the TME and leads to the emergence of innovative anticancer therapy [39,40,41]. Based on the cellular components of TME, we aim to summarize the current theories on the molecular mediators and pathogenesis of PNI and add the latest scientific research progress. The application of some new research methods and technologies may not only find new molecular targets for PNI but also provide valuable insights for the development of PNI in the tumor microenvironment and neural microenvironment; to obtain a better understanding of PNI may help to understand tumor metastasis and recurrence and will be beneficial for improving staging strategies, new treatment methods, and even paradigm shifts in our treatment of patients.

## 2. Main

Infiltrating cells constituting the tumor matrix include fibroblasts, bone marrow-derived cells (BMDCs), tumor-associated monocytes and macrophages, endothelial cells, endothelial progenitor cells (EPC), myeloid-derived suppressor cells (MDSCs), neurons, T regulatory cells (Treg), and pericytes, and different components in neuroimmune axis and many other unrelated pedigrees, and extra-cellular components (cytokines, growth factors, hormones, extracellular matrix, etc. (see Figure 1 and Table 1) [41,42]. Several populations of BMDCs are recruited into TME and niche, where they can differentiate into tumor-promoting populations, such as EPC, MDSCs, and macrophage-like cells [3]. Factors derived by cells in this microenvironment promote the migration and proliferation of adjacent vascular cells, tumor cells, and neurons, and grow together with tumors [43,44].

### 2.1. Monocytes/Macrophages

Monocytes/macrophages are usually the most abundant component in the immune cell population of the TME, and affect tumor progression and metastasis through interaction with tumor cells (see Figure 2) [45]. The main source of tumor-related macrophages (TAMs) is circulating mononuclear cells, which were derived from myeloid progenitors in the bone marrow and differentiated into mononuclear cells under the mobilization of the chemokine (C-C motif) ligand 2 (CCL2)/chemokine (C-X-C motif) receptor 2 (CCR2) axis and then enter the peripheral inflammation site through blood or further differentiate into macrophages in primary tumors [29,41,46], and they actively migrate to the tumorigenesis site and rapidly differentiate into TAMs during tumorigenesis, inhibiting various T-cell reactions, maintaining normal tissue homeostasis in response to various systemic infections and injuries, and helping anti-tumor escape [41]. They showed a high degree of functional plasticity, and can rapidly adapt to the disturbance of the environment around them.

TAMs were described into M1 and M2 subtypes according to their polarization state [47]. The M1 cell was activated by Th1 cytokine interferon-γ (IFN-γ), interleukin-12 (IL12), tumor necrosis factor (TNF), and microbial products to exert a tumor-killing effect, while the M2 cell was activated and differentiated by Th2 cytokines (such as IL4, IL5, IL10, IL13, colony-stimulating factor-1 (CSF1), transforming growth factor-1 (TFGβ1) and prostaglandin E2 (PGE2) to exert angiogenesis and immunosuppressive effect [41,43,45,48,49]. Anti-tumor M1 type was recruited at an early stage of tumor development, and the overexpression of the p50 subunit of nuclear factor kappaB (NFκB) in macrophages promoted the repolarization of M1 to M2 [42,50,51,52], gradually differentiated M1 into a tumor-promoting M2 type [53].

TAMs are generally considered to be M2 macrophages in the TME which, when exposed to hypoxia or lactate, secretes a variety of cytokines with metabolic functions, including IL6, TNF, CCL5, and CCL18 to enhance PNI [47]. TAMs express PD-L1 and can directly reduce the activation of T cells, then suppress the anticancer immune responses [42]. In addition, TAMs can promote tumor progression and invasion by up-regulating matrix metalloproteinases (MMPs) [54,55]. Carcinoma-produced MMP1 activates neuronal protease-activated receptor 1 (PAR1) and induces substance P (SP) release, activating carcinoma neurokinin 1 receptor (NK1R), which plays a pivotal role in tumor progression and PNI [42,56]. Macrophage-derived IL-1β induces non-neuronal cells to synthesize nerve growth factor (NGF), and activated macrophages can also release glial cell-derived neurotrophic factor (GDNF)-activated RET receptors to trigger tumor cell migration.

Evidence from clinical and experimental studies shows that there is a close relationship between TAM density and cancer cell metastasis in various cancers [28]. In PAAD, inflammatory monocytes (IM) expressing CCR2 can be driven by CCL2 released by Schwann cells at the PNI site to recruit preferentially to the nerve site, then differentiated into macrophages and enhance NI by expressing cathepsin B-mediated process to degrade the protein of nerve bundle membrane [29,39]. Protein species degraded by cathepsin B contained laminin, fibronectin, and collagen IV [29], which are important parts of the protective nerve bundle membrane, and its function as a peripheral nerve protective barrier is weakened when it is destroyed [57]. In addition, it has been reported that endoneurial macrophages (EMφ) can transform into microglia/macrophage subsets involved in cell defense and peripheral nerve regeneration [30]; EMφ around the nerve invaded by cancer is recruited to the tumor front in response to colony-stimulating factor -1(CSF-1) secreted by the tumor, then activated by tumor, secretes a higher level of GDNF as a chemical attractant of cancer cells, which induces phosphorylation of the transmembrane helper receptor RET in PAAD cells and activation of MAPK and PI3K pathways in downstream cancer cells of the extracellular signal-regulated kinase (ERK), promotes the migration to nerves of cancer cells, and enhances the PNI [30]. In addition, macrophages also regenerate injured nerves by secreting VEGF, which was also used by Schwann cells to guide the growth and migration of nerves for tumor axonogenesis [58].

### 2.2. Fibroblasts

Cancer-associated fibroblasts (CAFs) are an important subpopulation of fibroblasts to support tumorigenesis by stimulating angiogenesis, cancer cell proliferation, and mediating tumor-enhancing inflammation and invasion in tumors (see Figure 3) [13,59]. The origin of CAFs in TME has not been clarified, while in PDAC and CCA, they are considered to be differentiated from stellate cells [34,60], and bone marrow-derived circulating progenitor cells of hematopoietic or mesenchymal lineage were also reported to derive from CAFs [61,62]. Currently, it is believed that local tissue fibroblasts are activated into CAFs, which account for a large part of the stromal cells in TME and up to 80% of the tumor volume in advanced squamous cell carcinoma of the head and neck (HNSCC) [13,63], due to the influence of tumor-derived paracrine factors and cytokines [64].

They are responsible for the synthesis, deposition, and remodeling of most of the ECM in the tumor matrix and are thought to be sources of growth factors that affect paracrine, which affect the growth of cancer cells and have been shown to provide key signals supporting tumor progression and allow a small population of cancer cells to escape treatment [65,66,67,68]. Such as transforming growth factor β (TGF-β) secreted by CAFs induced a fibroblastic response in the surrounding environment to synthesize collagen, stiffen the tissue, and destroy microvascular structure by extensive collagen cross-linking, then the mechanical stress of the connective tissue collapses the nearby blood vessels to promote hypoxia [69]. Under hypoxic conditions, interleukin-6(IL-6) produced by cancer cells induces the activation of Schwann cells and triggers the development of PNI [70], which is also accompanied by the activation of a series of hypoxia-associated signaling pathways. In addition, a dense structure formed by fibrosis around a cluster of cancer cells in an adenocarcinoma caused internal high and lead to the rupture of glandular structure, simultaneously, the high-density fibers formed a track that contributed to the movement of the cancer cells to overflow and diffuse quickly [64]. By the anisotropic E-cadherin/N-cadher in junction, the frontier CAFs exert the physical force to enable the synergistic invasion of CAFs and cancer cells through a dual mechanism: CAFs pull cancer cells away from the tumor facilitating the invasion of cancer cells, while cancer cells polarize its migration away from the tumor to enhance the diffusion of CAFs [71,72]. In addition, fibroblasts contribute to Schwann-cell-induced axonal growth and, after nerve injury, ephrin-B on fibroblasts activates EphB2 receptors on Schwann cells, and fibroblasts through the ephrin-B/EphB2 signaling pathway induce Schwann cells to migrate through bridges as dense groups or cords to guide axons through injury site [73]. At the same time, CAFs can regulate the N-cadherin/β-catenin pathway through the production/secretion of the axonal guidance molecule Slit2, affecting the neural remodeling related to glandular catheter adenocarcinoma [74]. On the other hand, CAFs promote cancer invasion by secreting factors that cause cancer cell activation and matrix remodeling. Studies have shown that CAFs exist at the PNI site and the produced inactive matrix metalloproteinase-2(MMP-2) is activated by membrane-type 1 matrix metalloproteinase (MT1-MMP) produced by tumor cells, which degrades extracellular collagen in the perineural niche and promotes the spread of cancer cells in the perineural space [68,75].

### 2.3. Schwann Cells (SCs)

Schwann cells are the main cells of peripheral nerves, which wrap the single or multiple axons of peripheral nerves and promote signal transduction (see Figure 4). When it is damaged or invaded by tumor cells, the axonal injury may trigger the dedifferentiation and activation of SCs through various pathways [76]. The myelinated SCs are dedifferentiated into “repair SCs” (RSCs) [77] with a demyelinating phenotype. By producing a variety of neurotrophic factors and cell surface proteins, including GDNF, artemin and BDNF, p75 neurotrophic factor receptor (p75^NTR^), and N-cadherin, remodeling the matrix and releasing pro-inflammatory mediators to change the local signaling environment, the injured SCs can induce macrophages to synergistically remove myelin debris [77], and guide axonogenesis, participate in axon maintenance and post-injury repair, playing an important role in maintaining axon health and neuron survival, opening the way for subsequent nerve regeneration.

In the TME, SCs are thought to drive PNI—cancer cells using normal SCs nerve repair procedures to promote PNI [78]. SCs can directly interact with cancer cells through nerve cell adhesion molecule 1 (NCAM1) to break the connection between cells in the tumor cell cluster, thereby dispersing them into single cells [79]. Contact between SCs and cancer cells induces cancer cell protrusion and guides cancer cells to migrate from the cluster to SCs, thereby promoting the invasion and migration of cancer cells along the nerve [79]. L1 cell adhesion molecule (L1CAM) is overexpressed in SCs adjacent to cancer cells and invaded nerves, which strongly induces cancer cells as a strong chemoattractant by activating MAP kinase signaling, and L1CAM also up-regulates the expression of metalloproteinase -2 (MMP2) and MMP9 in PDAC cells by activating STAT3 and facilitates matrix remodeling along the axon [80,81]. MMPs secreted by SCs, especially MMP2 and MMP9, enhance the degradation of ECM and provide loose channels for the movement of cancer cells [82]. Schwann cells have also been identified as an important source of TGFβ, which can activate SMAD signals that induce migration, invasiveness, and PNI in PAAD cells, increasing the invasiveness of cancer cells [83]. Studies have suggested that before cancer cells migrate to peripheral neurons, SCs migrate to pancreatic or colon cancer cells through the NGF-TrkA-p75^NTR^ signaling pathway, but do not migrate to benign cells, so activated oncogenic SCs are likely to construct a pathway to cancer cells [84]. It has also been proposed in the literature that tumor cells activated SCs by c-Jun, while non-myelinating activated SCs form tumor-activated Schwann cell tracks (TASTs) as active scaffolds, and exert forces on cancer cells to enhance cancer mobility, promote cancer cell migration and invasion, a process similar to their reprogramming during nerve repair, leading to “neurogenesis” of precancerous cells and the periphery of tumor cells [85]. At the same time, SCs were demonstrated to be abundant in the surrounding stroma of the precancerous lesions of PAAD, possibly capable of recruiting immune cells at the PNI site, and macrophages recruited by SCs provided additional and persistent cytokine sources to further enhance neural invasion of tumor cells [20,77,84,86]. In a recent study, SCs activated by tumor produced prostaglandin E, polarized T cells to failure phenotype, leading to tumor-related immunosuppression, and play an important role in tumor–nerve crosstalk [87]. In addition, the differentiation of myoepithelial into SCs may be one of the mechanisms of PNI that occur in SACC^2^.

### 2.4. Neurons, Nerve Fibers, and Neurotransmitters

The interaction between nerve and cancer cells leads to the relationship of mutual growth promotion via the action of neurotrophic factors (NTFs) from nerve and cancer cells, and the pro-invasive and proliferative characteristics of autonomic neurotransmitters from nerve fibers [86,88].

Pathological neurogenesis is not only the physiological basis of chronic pain produced by the tumors but also contributes to the PNI [86,89,90,91]. Sensory nerve fibers undergoing pathological sprouting in cancers has been reported, which was driven by NTFs, especially the nerve growth factor (NGF), released from tumors and their associated stromal cells [92]. The expression of NTFs by cancer cells and nerves at the same time implies that cancer cells can make use of the same repertoire of trophic signals as nerves do to develop themselves [88]. In addition, after partial peripheral nerves were injured, the sustained demyelination state of the peripheral nerve relieves the axon of growth inhibition and encouraged nerve sprouting [93]. In a cancer such as PAAD, increased neural density (neural sprouting) and size (neural hypertrophy) occur, compared to normal pancreas innervation [86,94,95,96]. The mutual trophic interaction changes neural distribution in a solid tumor, which makes it more conducive to the exchange of signals between cancer cells and nerves [88]. In another way, it could be explained as the pathological neural plasticity induced by tumor-derived factors [86,91,97]. These signals were usually transferred via chemical substances with chemoattractive attributes such as NTFs, neuropeptides, neurotransmitters, and so on, leading to the mutual attraction between tumor cells and nerves, and enhancing cancer cells’ chemoattraction and motility. In addition, neurogenesis can also be induced by the recruitment of neural progenitor cells from the central nervous system (CNS), especially from the subventricular zone (SVZ), traveling through the bloodstream attracted by tumor-derived factors [42]. They will colonize in a tumor, and differentiate into functional autonomic neurons to produce adrenergic neurons and release neurotransmitters, and stimulate the growth of the tumor, mainly producing adrenergic neurons and releasing neurotransmitters to enrich the TME [98]. It was also proposed that cancer stem cells can differentiate and acquire an autonomic neuron-like phenotype, which may be able to perform neuron-related functions to influence the progression of tumors [42]. For example, cancer stem cells of gastric cancer and colorectal cancer can differentiate into sympathetic neurons producing tyrosine hydroxylase (TH) and parasympathetic neurons producing vesicular Ach transporter [99].

Studies have shown that in the process of PNI, when PAAD cells are close to neurons and SCs of peripheral nerves, they are attracted by neural components of peripheral nerves and migrate to neurons [84]. Neurotransmitters derived from fibers such as glutamic acid, aminobutyric acid (γ-aminobutyric acid), NE, or Ach stimulate the survival, proliferation, and migration of tumor cells [35]. Neurotransmitters regulate the immune-promoting and anti-immune responses and affect the TME through this indirect mechanism [34]. Sympathetic and parasympathetic nerve fibers release norepinephrine (NE) and acetylcholine (ACh) in tumors and lymphatic organs, as well as other neuromodulators, to reduce the anti-tumor immune response [100,101]. The sympathetic nerve fibers in the tumor are related to the early stage of cancer, and the angiogenesis switch is triggered by adrenergic signals [102]. In the late stage of tumor development, parasympathetic nerve fibers help to stimulate the invasion and metastasis of cancer cells [34,103,104]. NE derived from sympathetic nerve fibers activates β2- adrenergic receptors and activates through PKA/STAT3, which leads to the expression of NGF, MMP2, and MMP9 in PAAD cells, enabling them to migrate and invade and induce PNI [42]. NE can also stimulate the production of IL-6 and activate macrophages in TME, which can promote cancer migration and nerve invasion by releasing GDNF [30]. Parasympathetic nerve fibers release Ach, which inhibits immune response through nicotine receptors, while sensory nerve fibers release substances P and CGRP to activate mast cells and blood vessels [42]. Catecholamine activates the immunosuppressive switch in the TME of lung cancer, causing M1 to M2 macrophages to re-polarize and aggregate M2 polarized macrophages and MDSCs. At the same time, it reduces anti-tumor dendritic cells (DC), which leads to the synthesis and release of IL-10, inhibits immune response, leads to the synthesis and release of VEGF, and promotes angiogenesis [105]. In addition, another β2 Adrenergic-Neurotrophin (NT) loop driven by catecholamine can up-regulate NTs to increase sympathetic innervation and local NE accumulation [106].

### 2.5. Neurotrophic Factors, Neuregulins and Neuropeptides

Peripheral nerve microenvironment includes neurons, Schwann cells, and microglia/macrophages, which secrete various factors to participate in nerve homeostasis, dendritic growth, and axon germination [30].

Neurotrophic factors (NTFs) are a kind of protein, which play an important role in the development, survival, and apoptosis of neurons. Its members include nerve growth factor (NGF), brain-derived neurotrophic factor (BDNF), neurotrophin (NRTN), neurotrophins-3(NT-3), NT-4, NT-5, Artemin, etc, which promotes the growth of neurons and PNI together with receptors [100,107]. There are two different membrane protein receptors of NTFs, namely the tyrosine kinase receptor Trk(TrkA, TrkB and TrkC), which bind neurotrophins with higher affinity and specificity, and p75 neurotrophic receptor (p75^NTR^), which binds all neurotrophins with lower affinity and specificity [108]. NTFs interact with the extracellular domains of these two receptors, and transmit the signals related to the survival and apoptosis of nerve cells to the inside of cells, thus regulating the development and apoptosis of cells. When the nerve is subjected to transverse or focal crush injury, the expression of NGF and p75^NTR^ far away from the injured site is rapidly induced [84,107]. In oral squamous cell carcinoma (OSCC), Trk receptors have been confirmed to be overexpressed: Tropomyosin receptor kinase A (NGF), TrkB (BDNF, NT-4/5), and TrkC (neurotrophins 3, NT3) receptors, TrkC and its ligand NTF3 can promote the proliferation of SCs by inhibiting the formation of myeloid cells in the peripheral nervous system, and TrkC may also participate in PNI by regulating the interaction between SCs and tumor cells [68]. Ovarian cancer cells overexpress TrkB, and BDNF/TrkB can promote the migration and invasion of ovarian cancer cells by affecting myelination during nerve regeneration [67]. Macrophage-derived IL-1β induces non-neuronal cells to synthesize NGF, and tumor cells can also secrete NGF and BDNF to active their Trk receptors to stimulate nerve growth [109,110]. ACC, a neurotrophic tumor, and BDNF receptor, which promotes the survival and differentiation of axons and nerves, are abundant in the nerves invaded around the tumor. The normal prostate is one of the most abundant sources of NGF outside the nervous system [108]. Malignant prostate epithelial cells were reported to overexpress NGF and BDNF, but also the corresponding TrkB and TrkC receptors, which may be related to the migration of malignant cells, frequently occurs along nerves within the prostate, because it may provide abundant neurotrophins to act as chemoattractive guidance clues for tumor migration [111,112].

It was also reported that PAAD cells migrate preferentially toward human glial along a GDNF concentration gradient [113,114]. GDNF promotes the expression of integrin, activates MMP-9, and increases nuclear factor-κB(NF-κB), which affects nerve adhesion and invasion and promotes PNI [115]. By binding to the RET tyrosine kinase receptor, GDNF activates two downstream signaling pathways: the PI3-K-AKT pathway, which is involved in pro MMP-9 expression, and the RAS/RAF-MEK-1-ERK1/2 pathway, which is critical for the activation of MMP-9 [113]. In addition, neurons and their related SCs can also release soluble GFRα1 and GDNF(secreted by nerve macrophages), strongly activate RET in cancer cells and initiate the downstream activation of RAS/ERK, MAPK, JNK, and PI3-K-AKT signaling pathways, cancer cell migration, and PNI effect, and induced migration along the nerve [33,42,114,116,117].

Artemin sends signals through the Ret/GFRα3 receptor complex, as a member of the GDNF family of ligands [118]. In PAAD, over-expressed neurotrophin (NRTN) and Artemin activate RET tyrosine kinase (TK) by binding their homologous GDNF family receptor -α (GFRα) receptors, promoting cancer cell invasion and neuronal plasticity, and promoting the proliferation of nerve fibers around the tumor [119,120]. When the tumor volume increases, the pressure on nerve fibers increases and axon neurotrophic factors increase [89]. When an initial damage to the nerves by cancer cell, neurons and/or SCs produce Artemin/GFRα3 to repair the nerves, but the abundance of Artemin attracts further cancer cells to the site of injury, which produces a vicious cycle [121].

Axon injury may trigger the dedifferentiation and activation of SCs by releasing neuregulin (NRG) from the remaining neurons [20]. Neuregulin 1 (NRG1), which is the best studied of several neuregulin genes, can active ErbB receptor tyrosine kinases situated on glial membranes and signal to adjacent glial cells [122,123]. NRG1 forms is an EGF-like domain, binding to and activating receptors belonging to the EGF family of receptor tyrosine kinases erbB2, erbB3, and erbB4, after which it can produce erbB3/erbB2 or erbB4/erbB2 heterodimers or erbB4 homo-dimers and generate active signaling complexes and active an intracellular kinase domain present on erbB2 and erbB4, which phosphorylates specific tyrosine residues within the cytoplasmic tail of the receptor [124,125]. Then it leads to the activation of the mitogen-activated protein kinase (MAPK) and phosphatidylinositol-3-kinase (PI-3K) pathways [126,127,128]. A research work about PAAD has reported that nuclear FGFR1 regulates the transcription of NRG1, which generates an autocrine loop through ERBB2/4 to further drive invasion [129]. The expression and rapid release of NRG1 in neurons and their axons can be stimulated by NGF, BDNF, NT-3, and GDNF, which are extremely abundant in tumor microenvironment. NRG1 regulates the proliferation, migration and survival of developing SCs [130], which was one of the important members in PNI.

Tumor-derived factors and inflammatory mediators activate peripheral sensory fibers, resulting in the release of the substance P(SP), a neuropeptide that promotes tumor growth. SP enters the tumor, activates NK1R in cancer cells, and activates growth factor receptor through Src (EGFR, HER2) [3], and activates the MAPK pathway including extracellular signal-regulated kinases 1 and 2 (ERK1/2), to stimulate mitogenesis, induce cell proliferation, and avoid apoptosis by increasing the mRNA expression of MMP-2, MMP-9, VEGF, and VEGFR [131], which can also be produced by transactive EGFR and activation of NK-1R/Akt/NF-κB signaling pathway [132,133]. By these pathways, SP mediate the interaction between cancer cells and nerves, and promote the proliferation, invasion, and neurotropism of cancer cells, and PNI [132]. It was shown that SP promotes PAAD cell clusters gradually migrating to the dorsal root ganglions (DRGs) and SP-induced neurite regeneration extended to the clusters from the DRGs which provides an invasive pathway for the clusters [132,134].

In addition, neurogenic galanin (GAL) activates the G protein-coupled receptor galanin receptor 2 (GALR2) in tumors to induce NFATC2-mediated transcription of cyclooxygenase-2 and GAL, which causes the crosstalk between nerves and cancer cells [14]. Prostaglandin E2 promotes cancer invasion by promoting the secretion of pro-inflammatory mediators and neuropeptides by tumor cells, GAL released by cancer induces neuritogenesis, facilitating PNI [14,22]. In addition, GALR2-RAP1-p38MAPK-mediated inactivation of Tristetraprolin(TTP), an RNA-binding protein that promotes decay of transcripts of proangiogenic factors (including IL-6, VEGF, IL-8) [135,136,137], and then induces angiogenesis, which facilitates tumor progression by supplying oxygen and nutrients [138].

### 2.6. Chemokines

Chemokines play an important role in cell–cell interaction, which may mediate the chemical attraction of neurons and/or SCs to cancer cells. Infiltration of tumor cells into nerves can lead to nerve injury and release CCL, which induces inflammatory reaction of nerve repair, and then induces the migration of cancer cells expressing CCR to injured nerves, and finally promotes the PNI effect [139].

In prostate cancer (PRAD) cells, the expression of CCR2 (the receptor of CCL2) promotes NI, and the expression of CCR2 is closely related to the activity of MAPK and Akt pathways and the migration of cancer cells to chemokine (C-C motif) ligand 2 (CCL2) and DRG [140]. In PAAD, sensory-neuron-derived mediators CXCL10 and CCL21 pass through complementary receptors CXCR3 and CCR7 on tumor cells, activating AKT, MEK, and RAC signaling pathways in tumor cells to mediate migration [141]. CXCL12 derived from PAAD cells can induce SCs to infiltrate the tumor in the early canceration process and promote cancer cells to attract and migrate to the nerves [141,142]. In addition, the main chemokines include chemokine CCL2 and matrix-derived factor 1 (SDF-1/CXCL12), which induce cancer cell migration under the action of CCR2 and CXCR4 receptors respectively, and can also recruit bone marrow-derived cells (BMDC) and M2 macrophages, and the recruited macrophages secrete GDNF, which can activate RET-GDNF receptor α1 (GFRα1) in cancer cells to promote PNI and the invasion of cancer cells [30,143]. Nerve-related macrophages accumulate in the nerves invaded by tumor cells along the gradient of CCL2 and CSF-1 recognized by CCR2 and CSF-1R receptors, respectively [29,144]. Nerve-derived C-X3-C motif chemokine ligand 1 (CX3CL1) and NT-3 further support the interaction between nerve tumors [145], and the former enhances the adhesion between cancer cells and nerves, while the latter regulates the interaction between SCs and cancer cells [3].

### 2.7. Semaphorin

As a family of membrane-related or secreted glycoproteins, it was reported semaphorin can participate in axon guidance and regulate cell migration such as WBCs, neurons, and endothelial cells, to attend cancer progression [146]. Axonal guidance was the most important function, Semaphorin family can mediate axonal guidance by combining plexin family members, such as Sema3D and plexin D1 (PlxnD1) [147], and semaphorin-3A (Sema3A) binding to plexin A1 [148], and the tripartite complexes formed by semaphorins-3 (Sema3s), plexin receptor and neuropilin coreceptor can also mediate axon guidance and invasion [148,149,150]. Besides, there still are some other pathological processes mediated by this combination of the Semaphorin family and plexin family, such as that semaphorin-4D (Sema4D) induce tumor angiogenesis and vascular maturation by binding to the plexin B1 receptor on endothelial cells [42]. They can also have an effect on regulating tumor cell survival, such as Sema3E-PlxnD1 signaling suppressing apoptosis in breast cancer [151]. Over-expression of Semaphorin-4F (Sema4F) in PRAD cells contributes to the communication between nerve fibers and cancer cells and induces the proliferation and migration of PRAD cells [152].

### 2.8. Tumor Microenvironment

In solid tumors, the rapid growth of tumor tissue, high expansion, and incomplete vascular system in tumor tissue, will lead to insufficient oxygen supply in tumor tissue, and the TME presents overall hypoxia [153,154]. Due to hypoxia, tumor cells can only metabolize energy through anaerobic glycolysis, which will lead to the accumulation of lactic acid [155,156]. At the same time, ion-exchange proteins on tumor cell membranes are constantly transporting H+ inside cells to outside cells to avoid self-acidosis [157]. These cellular reactions also caused the PH of the tumor microenvironment to decrease to different degrees, and the overall environment was acidic [158]. In the microenvironment of tumor occurrence and development, hypoxia, and acidity, a lot of apoptosis will occur in tumor tissues and peripheral tissues, releasing cell fragments and chemokines, leading to infiltration of inflammatory cells and secretion of inflammatory factors [158]. At the same time, the occurrence and development of the tumor itself will also trigger the immune response of the immune system, causing inflammatory cells to gather in this area, and triggering a severe inflammatory response [159]. This local microenvironment tumor promotes the growth and infiltration of tumor cells into nerve tissue [3,29]. It interacts with the perineural environment (nerve cells, glial cells, and their products) to further change the microenvironment and promote PNI [3].

Mitochondrial dysfunction and altered glucose metabolism are considered the typical changes in the tumor microenvironment, and this metabolic change is potentially related to the etiology of nervous system degeneration and nerve injury [160,161]. Oxidative stress was reported to induce chronic neuroinflammation, which contributes to the functionality switch of astrocytes from neurotrophic to neurotoxic, which can release more lactate to reinforce their energetic support to neurons, although upregulating the detrimental pathway [162]. An important feature of energy metabolism in tumor cells is called the “Warburg effect”, which is characterized by heavy dependence on glycolysis and the production of a large amount of lactate [163]. Lactate is easily absorbed and decomposed by tumor cells to produce energy [164]. Lactate can promote tumor invasion and metastasis, play an immunosuppressive role, and promote tumor development by inducing and recruiting immunosuppressive-related cells and molecules [165,166]. In particular, lactate is abundant in the nervous system, which was provided by myelinating Schwann cells (mSCs) using aerobic glycolysis to support action potentials propagation along axons [167,168]. In addition, glycogen is indicated to be present in the peripheral nerve, primarily in mSCs [168]. We speculate that this is helpful for the tumor to invade the nerve and obtain a faster spread speed after the tumor invades the nerve.

## 3. Single-Cell Spatial Transcriptomics (sc-ST)

Recently, single-cell RNA sequencing (scRNA-seq) has provided unprecedented resolution for revealing complex cellular events and deepening our understanding of biological systems [169,170]. However, most scRNA-seq protocols require the complete recovery of cells from tissues and the guarantee of cell survival, which excludes many cell types from the scope of research, and largely destroys the spatial background that could have provided information for cell identity and function analysis. However, the functions of many biological systems, such as embryos, liver lobules, intestinal villi, and tumors, depend on the spatial organization of their cells. In the past decade, high-throughput technology has been developed to quantify gene expression in space, and computational methods can be used to identify genes with spatial patterns and describe the neighborhood in tissues, which is called “spatial transcriptomics (ST)”. The emergence of this new technology has improved the spatial resolution and high-dimensional evaluation of gene transcription [171]. Some researchers have used single-cell spatial transcript information (sc-ST) obtained by combining single-cell sequence information with spatial transcriptome technology to decipher cell components in the tumor invasion niche, and its transcription reprogramming and potential crosstalk, so that the resolution of research becomes higher and accurate spatial positioning can be obtained. It is well known that hypoxia, EMT, and inflammation signatures contributed to intra-tumor spatial variations, which led to functional differences in different niches [172]. In a study of skin cancer, the author confirmed that tumor cells showed a collective migration phenotype and strongly expressed cytokine A, which contributed to the spatial organization of the invasive niche of basal cell carcinoma [5]. ST showed that tumor cell subgroups in the hypoxia group changed, and different subgroups showed their location characteristics and different gene markers. Subgroups at the front of invasion showed higher proliferation ability, invasiveness, and response to stress under hypoxia [155]. This tech has been also used to explore the spatial landscape of multiple cell subpopulations in esophageal squamous cell carcinoma (ESCC). A study reported the intra- and inter-tumoral heterogeneity of ESCC, which, exploring inflammatory fibroblasts (iCAFs), were mostly clustered in the stromal regions, whereas no difference was found in the distribution of myofibroblast (myCAFs) between cancer and stromal regions [173]. Some specific pathways enriched in iCAF subpopulations may be candidates for future research in the progression of ESCC. Another scRNA-seq analysis study also reported the difference in cells between the primary and metastatic sites of HNSCC. Previous studies have also confirmed that the position of macrophages relative to tumor cells is different in various characteristics [45]. In an experiment in diffuse gastric cancer, fibroblasts, endothelial cells, and bone marrow cells were found to be enriched in the deep layer, and it was found that cell-type-specific clustering further revealed that the transition from the shallow layer to the deep layer was related to the up-regulated enrichment of CCL2 transcript in inflammatory endothelial cells and fibroblasts [40]. ST analysis confirmed that stromal cell clusters located at the front of the tumor invasion were identified, which expressed genes related to hypoxia signal transduction, angiogenesis, and cell migration, which proved that hypoxia signal was involved in the metastasis process of invasive gastric cancer [174]. Another study also testified to this view by sc-ST technique, which showed that tumor cells at the outermost edge responded strongest to their local microenvironment, behaved most invasively, and activated the process of epithelial-to-mesenchymal transition (EMT) to migrate to low-confluence areas, and induced similar phenotypic plasticity in neighboring regions [175]. In a study of oral squamous cell carcinoma (OSCC), the spatial localization of nerves in TME was evaluated and successfully summarized into four types of PNI to provide more detailed and accurate pathological information [38]. For the exploration of the interaction between nerve and tumor, the conventional laboratory methods are limited; we could only observe an averaged biological signal over a large number of cells or conventional description based on morphological layer, so the sc-ST should be used to explain the communication between nerve and tumor cells and provide more exact information of tumor cells and nerve [176]. In future research, we expect to make full use of sc-ST to further explain the phenomenon of a tumor’s perineural infiltration, observe its interaction at different spatial points, and explore its mechanism.

## 4. Conclusions and Future Perceptives

A lot of early studies have been carried out to explore the inner pathology process of the perineural invasion of some cancers. Important cells, such as SCs, TAMs, CAFs, and some other related cells have been thoroughly studied or are on the way. However, all of these studies were limited and it appears we cannot go any further at present. In this review, we try to introduce a new technology into this pathology process, single-cell spatial transcriptomics. This technology has been used in some studies of common tumors; we can observe the cell components in tumors and their distribution, and then explore the information that we had no chance to capture before through conventional research techniques. According to some studies that have been reported, we hope we can find some transforming cells with an important effect on the PNI process, such as some special CAFs explored in gastrointestinal cancer and the cell-type-specific clustering characteristics. We are trying to understand PNI with this new technology. Maybe soon, we can find a way to stop PNI and the distant metastases that follow.

## Figures and Tables

**Figure 1 cancers-15-01360-f001:**
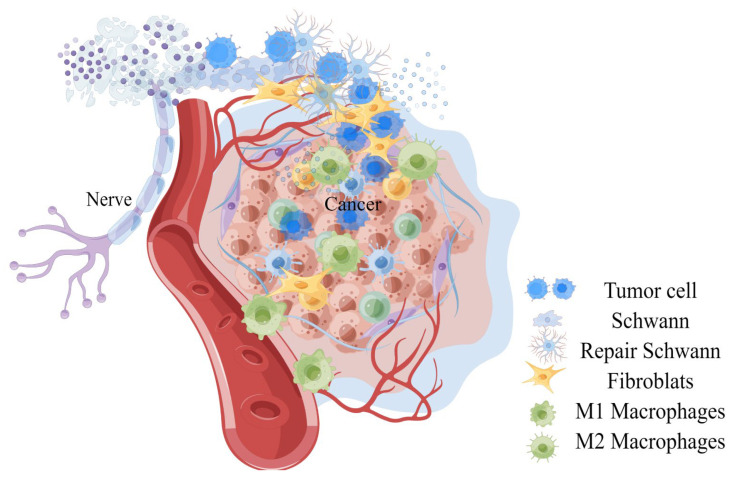
A general observation on the invasion of peripheral nerves of tumors.

**Figure 2 cancers-15-01360-f002:**
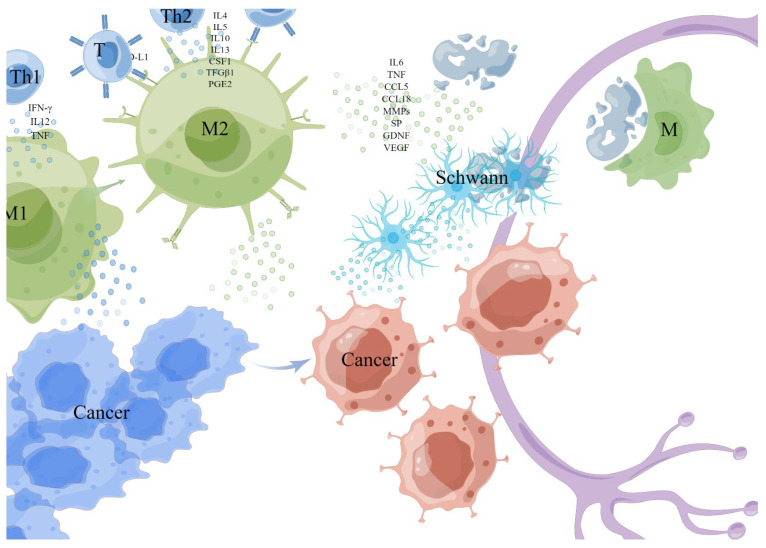
Monocytes/macrophages. The M1 cell was activated by Th1 cytokine interferon-γ (IFN-γ), interleukin-12 (IL12), tumor necrosis factor (TNF), and microbial products, while the M2 cell was activated and differentiated by Th2 cytokines (such as IL4, IL5, IL10, IL13, colony-stimulating factor-1 (CSF1), transforming growth factor-1 (TFGβ1) and prostaglandin E2 (PGE2). TAMs (generally considered to be M2) exposed to hypoxia or lactate secretes a variety of cytokines with metabolic functions, including IL6, TNF, CCL5, and CCL18 to enhance PNI. TAMs express PD-L1 and can directly reduce the activation of T cells, then suppress the anticancer immune responses. TAMs promote tumor progression and invasion by up-regulating matrix metalloproteinases (MMPs). TAMs degrade the protein of nerve bundle membrane by expressing cathepsin B-mediated process. This picture is drawn by Figdraw (www.figdraw.com).

**Figure 3 cancers-15-01360-f003:**
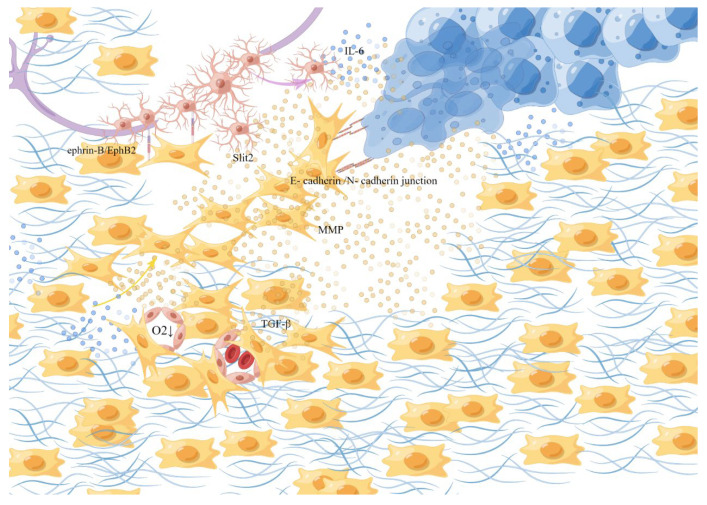
Fibroblasts. Local tissue fibroblasts are activated into CAFs, due to the influence of tumor-derived paracrine factors and cytokines. They help with the synthesis, deposition, and remodeling of most of the ECM in the tumor matrix. CAFs secrete TGF-β to synthesize collagen, stiffen the tissue, destroy the microvascular structure, and collapse the nearby blood vessels to promote hypoxia, form a track contributing to the movement, overflowing, and diffusion and of the cancer cells, and exert physical force by the anisotropic E-cadherin/N-cadherin in junction, degrade extracellular collagen and induce migration of SCs through the ephrin-B/EphB2 signaling pathway. This picture is drawn by Figdraw (www.figdraw.com).

**Figure 4 cancers-15-01360-f004:**
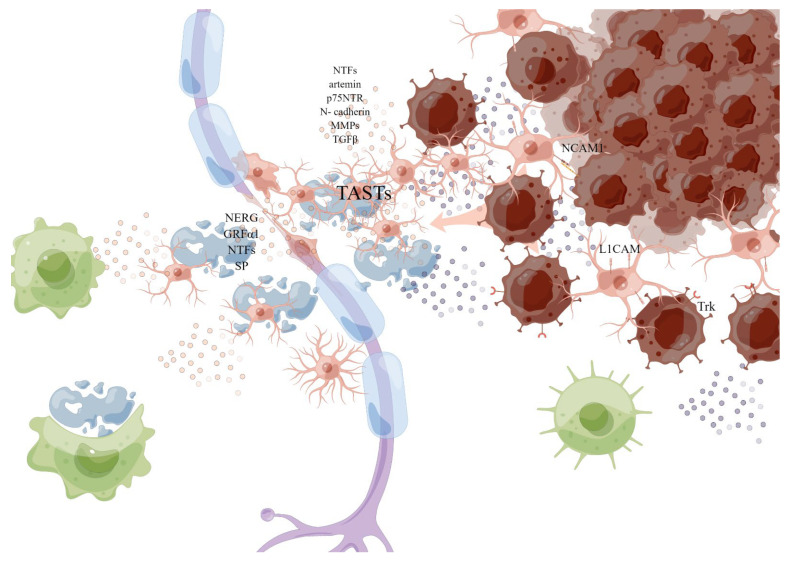
Schwann cell. The myelinated SCs are dedifferentiated into “repair SCs” (rSCs) with a demyelinating phenotype when it is damaged or invaded. By producing a variety of neurotrophic factors and cell surface proteins to interact with cancer cells, including GDNF, artemin and BDNF, p75 neurotrophic factor receptor (p75NTR), TGFβ, and N-cadherin, remodeling the matrix to help the migration of cancer cells, and releasing pro-inflammatory mediators: TGFβ, prostaglandin E, to change the local signaling environment, the injured SCs can induce macrophages to synergistically remove myelin debris, and produce (TASTs) as active scaffolds to guide axonogenesis or cancer invasion. This picture is drawn by Figdraw (www.figdraw.com).

**Table 1 cancers-15-01360-t001:** Summary of factor function and its mechanism.

Factor Family	Factors	Receptor	Mechanisms	Functions
Neurotrophic factors	NGF	TrkA	Activation of Trk receptors results in either neuronal differentiation or mitogenic stimuli. Transmit the signals related to the survival and apoptosis of nerve cells to the inside of cells, thus regulating the development and apoptosis of cells.	Stimulate nerve growth
	BDNF, NT-4/5	TrkB		Affects myelination during nerve regeneration; Promote migration and invasion
	NT3	TrkC		Inhibit the formation of myeloid cells in the peripheral nervous system to promote the proliferation of SCs
	NRTN/Artemin	GFRα	Activate RET tyrosine kinase (TK) by binding their homologous GDNF family receptor -α (GFRα) receptors	Promote cancer cell invasion and neuronal plasticity; Promote the proliferation of nerve fibers around the tumor; Regulate the interaction between SCs and tumor cells.
	GDNF	GFRα1	Active RAS/ERK, MAPK, JNK, and PI3-K-Akt.	Prompt pro-MMP-9 expression and activation of MMP-9 to affect nerve adhesion and invasion; Initiate cancer cell migration, and PNI effect, and induced migration along the nerve; Prompt invasion and metastasis formation; Enhance the expression of integrin.
Neuregulins	NRG1	ErbB	Active MAPK, PI-3K.	Increase the gap connection communication between SCs; Adjust the physiological characteristics of SCs, and promote the movement and migration of SCS
Neuropeptides	SP	NK-1R, EGFR, HER2	Active MAPK (including ERK1/2 and P38mapk); Active NK-1R/Akt/NF-κB signal pathway; Transactive EGFR and HER2.	Increase MMP-2, MMP-9, VEGF, and VEGFR; Stimulate cell proliferation; Lead to growth; Avoid apoptosis.
	GAL	GALR2	Active MAPK signal pathways and inactive TTP.	Promote Prostaglandin E2 generation to promote the secretion of pro-inflammatory mediators and neuropeptides by tumor cells; Promotes cytokine secretion (including IL-6, VEGF, IL-8); Induce angiogenesis and neurogenesis.
Chemokines	CXCL10, CCL21	CXCR3, CCR7	Active AKT, MEK, and RAC signal pathways in tumor cells	Promote cancer cells’ invasiveness, migration, proliferation, epithelial–mesenchymal transition; and sensitize sensory nerves; Recruit bone marrow-derived cells (BMDC) and M2 macrophages; GDNF secreted by the recruited macrophages activates RET-GDNF receptor α1 (GFRα1) in cancer cells and promotes the invasion of PNI and cancer cells; Enhance the adhesion between cancer cells and nerves.
	CCL2	CCR2	Active MAPK, AKT signal pathways.	
	CXCL12	CXCR4/CXCR7	Active AKT, ERK, and sonic hedgehog-dependent pathways;	
	CCL2	CCR2	Active RET-GDNF receptor α1 (GFRα1) in cancer cells.	
	CX3CL1	CX3CR1	CX3CL1 direct contact CX3CR1 to adhere to nerve cells.	

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
