# Peer review of "Important Cells and Factors from Tumor Microenvironment Participated in Perineural Invasion"

_cancers, 2023, doi:10.3390/cancers15051360_

Round 1

Reviewer 1 Report

The review of Chen et al is devoted to very interesting topic on “Important Cells and Factors from Tumor Microenvironment Participated in Perineural Invasion”. Indeed, this topic attracts a lot of attention because it describes one more type of the tumor invasion process.

However, despite a good volume of data presented in the review, it is very difficult to understand this topic based on the corresponding paper. I encourage authors to improve their paper by

Main points:

1.     The text should be divided into paragraphs in a logic way. It is not possible to read text that is not logically divided into paragraphs. For instance, chapter 2.4 – a lot of different bulked information.

2.     In “Background”: More comprehensive description of PNI should be given, what morphological features does it have, what is known about the functional role of interactions between nerves and cancer cells? What is neoneurogenesis? What are direct or indirect effects of nerves on tumors? What about effect of cancer cells on nerves?

Neoneurogenesis

3.     The figures are very beautiful but carry a low information load. How produce molecular factors indicated? What are the consequences? The reader should only guess. Al least, the comprehensive description of each of the figures should be placed.

4.     Only a superficial description of molecular factors taking part in PNI is provided. Please, give the examples in more details to understand the molecular consequences of them for tumorigenesis.

5.     All-in-all, this is difficult to give the insight about PNI based on the review. I suggest to place a table where authors will summarize data on molecular factors taking part in PNI and corresponding certain examples with the functional consequences.

6.     Chapter 2.8. What is known about the role of exosomes in PNI?

7.     How can we appropriately target the nerves and PNI to prevent the tumor progression? How our knowledge about this can help us to improve therapy? The additional chapter is required.

Author Response

Dear reviewer,

Thank you for your advice.

I have revised this paper. I have revised this article. I have also answered your questions one by one below. I hope you think this is OK.

Zirong Chen

09/02/2023

Main points

The text should be divided into paragraphs in a logical way. It is not possible to read text that is not logically divided into paragraphs. For instance. in chapter 2.4 - a lot of different bulked information

Chen: I have divided this into paragraphs in a logical way. I hope this segmentation will help readers understand the article.

In“Background”. A more comprehensive description of PNI should be given. what morphological features does it have. what is known about the functional role of interactions between nerves and cancer cells? What is neoneuroenesis? What are the direct or indirect effects of nerves on tumors? What about the effect of cancer cells on nerves?

Chen: The explanation about these questions I have added to Chapter 2.4  and other contents about the description of PNI were added to the Background part.

Neoneurogenesis

  1. The figures are very beautiful but carry a low information load. How are produced molecular factors indicated? What are the consequences? The reader should only guess. At least, a comprehensive description of each of the figures should be placed.

Chen: The explanation about the mechanism of the factor has been added in detail and a description of the picture.

  1. Only a superficial description of molecular factors taking part in PNI is provided please, give examples in more detail to understand the molecular consequences of them for tumorigenesis.

Chen: Relative information has been added in chapter 2.4. and chapter 2.5.

  1. this is difficult to give insight into PNI based on the review. I suggest placing a table where authors will summarize data on molecular factors taking part in PNI and corresponding certain examples with the functional consequences.

Chen: The table has been listed.

  1. Chapter 2.8. What is known about the role of exosomes in PNI?

Chen:  Exosomes (EVs) palys a role as a transporter. The contents carried by exosomes derived from cancer cells can internalize into receptor cells and regulate their biological functions and signal transduction, enhance the crosstalk between tumors and promote tumor innervation. The most important function of it was the transportation function, and the contents of EVs were the key to making effects. After the article was rearranged and revised, we decided to delete this paragraph.

  1. How can we appropriately target the nerves and PNI to prevent tumor progression? How our knowledge about this can help us to improve therapy? An additional chapter is required.

Chen: the title of this paper was “Important Cells and Factors from Tumor Microenvironment Participated in Perineural Invasion.” So, I prefer to describe how this happened. If I want to introduce the treatment of PNI in detail, I think the length of the article will far exceed my expectation, so I don't want to add this part. I hope you can understand my dilemma. Please contact me again if you think this part is really important.

Reviewer 2 Report

In the Review “Important Cells and Factors from Tumor Microenvironment 2 Participated in Perineural Invasion. by Chen et al., has explained the effect of tumor and microenvironment factors in perineural invasion of solid tumors.  The paper is well written; however, I suggest few minor changes.

1. Please include a future prospective section in the review.

2. Please include a small part how the metabolomics of the tumor cells or tumor microenvironment cells can affect the perineural invasion.

Author Response

Dear reviewer,

Thank you for your suggestion.

  1. I have added the future prospective section in the last paragraph.
  2. I have added a small part about how the metabolomics of the tumor cells or TME cells can affect the PNI.

Sincerely,

Zirong Chen

01/02/2023

Round 2

Reviewer 1 Report

The authors have addressed all issues requierd.